# The Impact of Physical Activity on Depressive Symptoms among Urban and Rural Older Adults: Empirical Study Based on the 2018 CHARLS Database

**DOI:** 10.3390/bs13100864

**Published:** 2023-10-21

**Authors:** Xueyu Jin, Huasen Liu, Eksiri Niyomsilp

**Affiliations:** 1Taiyuan Institute of Technology, Taiyuan 030008, China; jinxueyu@tit.edu.cn; 2School of Sports and Leisure, Xi’an Physical Education University, Xi’an 710068, China; 105025@tea.xaipe.edu.cn; 3School of Management, Shinawatra University, Pathum Thani 12160, Thailand

**Keywords:** depressive symptoms, physical activity, older adults, urban–rural differences

## Abstract

Using data from the China Health and Retirement Longitudinal Study 2018, we employed the propensity score matching method to examine the effect of physical activity on depressive symptoms among older adults across rural and urban areas. The study sample consisted of 5055 participants, with urban and rural populations representing 31.3% and 68.7%, respectively. This study found that rural older adult individuals exhibited a greater incidence of depressive symptoms and lower physical activity levels when compared to their urban counterparts. Engagement in high-intensity physical activity was identified as an effective method for mitigating depressive symptoms among older adults. However, the moderating effects of physical activity were only observed among urban older adult individuals. Our findings revealed a cross-sectional correlation between physical activity and depressive symptoms among older adult Chinese individuals, and this link differed between urban and rural areas. Although high-intensity physical activity has a positive effect on depressive symptoms among older adults, physical activity interventions should sufficiently consider the variations in older adults’ living conditions and environments due to urban–rural differences so that interventions can be customized to improve the mental health of older adults.

## 1. Introduction

The findings of the seventh national population census conducted in 2010 revealed that there are 264,018,766 individuals in China aged 60 and above, making up 18.70% of the total population. Within this group, 19,063,528,280 individuals are aged 65 and over, which accounts for 13.50% of the entire population. In comparison to the sixth national census conducted in 2010, the growth rate of the population aged 60 years and over was as high as 5.44% [1]. These figures suggest that China is transitioning from a mildly to moderately aging society. Although aging increases the level of social security expenditure, a faster aging rate hinders this rise in expenditure [2]. Furthermore, the increase in the aging population poses various socio-economic issues, particularly the need for medical care and security, thereby placing a significant burden on individuals, families, and society. Research indicates that an increase of 1% in the healthcare burden of older adults results in a decrease of 0.083% in the GDP growth rate [3]. As the nation with the world’s largest population of older adult individuals, China’s demographic shift towards aging before achieving prosperity is placing increased demands on public healthcare services, as well as economic progress. Subsequently, the execution of healthy aging strategies throughout one’s life course has garnered interest across society. It is strategically important for proactive health management and disease prevention strategies applied to older persons to improve quality of life and ameliorate the economic and social predicament of aging.

### 1.1. Symptoms of Depression among Older Adults

Depression is a prevalent mental disorder amongst older adults, and it is predicted to become the world’s leading disease in terms of burden by 2030 [4]. China is dealing with a significant public health issue due to the gradual rise in the number of depressed older adults [5]. The data show that depression and depressive symptoms have a significant impact on the healthcare costs of older adults in rural China, accounting for about 47.26% of the total expected individual healthcare expenditure [6]. As older adults age, living alone and becoming isolated can exacerbate depression and lead to depressive symptoms [7]. Evidence suggests that depression among older adults has a detrimental impact on chronic diseases, like cardiovascular and circulatory diseases, as well as endocrine metabolic diseases [8]. Moreover, depression is a significant risk factor for non-suicide deaths among older adults [9]. Furthermore, it has been posited that depression is not easily predictable among older adults, often presenting as long-term and/or recurrent episodes that may be challenging to treat [10]. Nevertheless, the dynamic transition of depression offers the possibility of preventing and reversing depression among older adults, and health-related behaviors are one of the most effective ways of preventing and ameliorating depressive moods [11,12].

### 1.2. Physical Activity and Depressive Symptoms

Physical activity, as an important form of active health management and disease prevention for older adults [13], can reduce depressive symptoms through a range of biological and psychosocial mechanisms, such as stimulating neuroplasticity, reducing inflammation, and enhancing social support and co-efficacy [14]. Previous investigations have generally confirmed the favorable effects of physical exercise on the alleviation of depressive symptoms [15,16]. However, evidence suggests that the level of physical activity is not associated with the risk of depressive symptoms among older adults [17]. Additionally, the dose–effect relationship between physical activity and depressive symptoms requires further exploration [18]. Wang et al. [19] discovered a negative association between almost-daily exercise and depressive symptoms among older adults. Therefore, the effect of physical activity on depressive symptoms among older adults in China is worth investigating. This study focuses on the strength of the positive impact of mitigating depressive symptoms among older individuals and the differences between residents of urban and rural areas.

### 1.3. The Present Study

To decrease depression prevalence and promote active health management among older adults, in this study, we employed a multiple linear regression model using data from the China Health and Retirement Longitudinal Study (CHARLS) to examine the relationship between physical activity and depression risk among older adults while considering urban–rural disparities. The following hypothesis was tested.
**Hypothesis:** High physical activity reduces the risk of depression among older adults.

## 2. Materials and Methods

### 2.1. Data Source

The data used herein come from the CHARLS 2018 survey, a large-scale social survey program conducted in China. The China Health and Retirement Longitudinal Study (CHARLS) is an interdisciplinary survey project carried out by Peking University and funded by the National Natural Science Foundation of China. CHARLS covers 150 counties and 450 villages in China and is conducted biennially. The phase of the CHARLS project analyzed herein used multistage sampling, with probability-proportional-to-size (PPS) sampling methods, at both the district and village levels. At the individual level, the survey was conducted using a filtered questionnaire in which one household member aged over 45 years was randomly selected as the primary respondent in each sample household, and they and their spouse were interviewed. This survey comprised essential personal details, household composition, financial assistance, physical wellbeing, measurements, healthcare utilization, and insurance coverage [20]. The study participants comprised members of the geriatric population, with data being extracted solely from individuals aged 60 to 90 years. After matching personal IDs and excluding data samples lacking necessary information for the research, we obtained a total of 5055 samples. Of these samples, 51.0% were male and 49.0% were female. The urban population accounted for 31.3%, and the rural population accounted for 68.7%.

### 2.2. Variable Selection

#### 2.2.1. Depressive Symptoms

In this survey, the CESD-10 scale was used to investigate the risk of depression among older adults. The 10 items refer to how one felt and behaved during the last week: (1) I was bothered by things that do not usually bother me; (2) I had trouble keeping my mind on what I was engaged in; (3) I felt depressed; (4) I felt that everything I achieved was acquired through effort; (5) I felt hopeful about the future; (6) I felt fearful; (7) I slept restlessly; (8) I was happy; (9) I felt lonely; and (10) I could not get “going”. Every item has the same selective answers: rarely or none of the time, sometimes, occasionally, and most or all the time. The questionnaire has a score distribution of 0–30, with scores of 10 and above defined as the presence of depressive symptoms. The scale has been shown to have high validity and reliability in surveys of older adults in China [21].

#### 2.2.2. Physical Activity Level

The CHARLS follow-up interview questionnaire collected data on the quantity and duration of highly physically demanding activities (such as lifting heavy objects, ploughing, aerobic exercise, etc.), moderately vigorous physical activities (such as lifting light objects, tai chi, sprinting, etc.), and light physical activities (such as walking) carried out by the respondents during the previous week [22]. Firstly, using the metabolic equivalent (MET) assignment of each physical activity in the IPAQ (International Physical Activity Questionnaire) short version, we estimated the respondents’ energy expenditure regarding physical activity over one week. The MET values for light-intensity, moderate-intensity, and high-intensity physical activity were 3.3, 4.0, and 8.0, respectively. The intensity level of physical activity is determined by multiplying the MET value assigned to the particular activity type by the frequency per week (d/w) and amount of time per day (min/d). The sum of these three intensity levels represents the total physical activity level. Physical activity was classified into three categories based on the IPAQ rubric [23]: low-intensity physical activity (<600 METs/min per week), moderate-intensity physical activity (600–3000 METs/min per week), and high-intensity physical activity (>3000 METs/min per week). To enable comparative analyses with international studies on physical activity levels, two categories of physical activity levels were established: low intensity and moderate-to-high intensity.

#### 2.2.3. Covariates

In this study, the covariates comprise the foremost pertinent factors impacting depression levels and physical activity among older adults. These covariates consist of fundamental characteristics such as age, gender, level of education, area of residence, and marital status; health status, including self-reported health, chronic disease type, and disability status; health care and insurance status concerning medical care habits and expenses and participation in various forms of medical insurance; and actual income status in the past year. The survey’s precise variables are elucidated in the “China Health and Retirement Longitudinal Study Wave 4 (2018) Questionnaire”, which is appended in the Appendix A.

#### 2.2.4. Statistical Analysis

The dataset was acquired via a DTA-formatted database application and analyzed utilizing STATA/SE17.0. A primary descriptive analysis of the data features was carried out beforehand. Categorical variables were formatted as n (%), and continuous variables were formatted as means (M) ± standard deviation (SD). We proceeded to analyze the factors affecting the physical activity levels of older adults through multifactor logistic regression analysis, and the Average Marginal Effects (AME) of each variable across all samples were calculated. Propensity score matching was then utilized to assess the effect of physical activity on depression levels among older adults, effectively eliminating bias in the observed covariates and sample endogeneity [24]. The specific steps taken were as follows.

In the initial stage, the propensity scores for two physical activity levels, namely, low and moderate-to-high, were matched among older adults:$$\ln\left[\frac{P_{i}}{1-P_{i}}\right]=\beta_{0}+X_{i}\beta_{i}+\varepsilon$$
where *i* represents an individual older adult. The probability with respect to the physical activity level of older adults is represented by $P_{i}=P\left(D_{i}=1/X_{i}\right)$, where $X_{i}$ denotes the influencing factors of the physical activity level of older adults, encompassing their individual characteristics, family characteristics, health, and income levels. The coefficients of the influencing factors are represented by $\beta_{i}$, with $\beta_{0}$ being a constant term, and ε is a random perturbation term.

In the second stage, an estimated counterfactual was calculated for older individuals exhibiting moderate-to-high physical activity levels, using propensity score values to determine the average treatment effect for the treatment group.
$$ATT=E\left(\frac{Y_{i}\left(1\right)}{D_{1}}=1,P\left(X_{i}\right)\right)-E\left(\frac{Y_{i}\left(0\right)}{D_{i}}=0,P\left(X_{i}\right)\right)$$

Above, $Y_{i}\left(1\right)$ represents the status of depression among older adults with moderate-to-high levels of physical activity, and $Y_{i}\left(0\right)$ depicts the status of older adults with low levels of physical activity. $D_{i}$ is a dummy variable that categorizes older adults based on their level of physical activity, and $P\left(X_{i}\right)$ is an individual’s propensity score.

This study assessed the average treatment effect of physical activity levels on depressive symptoms among older adults by utilizing three matching methods: nearest neighbor, kernel, and caliper matching. The aim was to test for robustness.

## 3. Results

### 3.1. Descriptive Characteristics of the Sample

The mean CESD-10 score for the final sample was 19.17. The mean CESD-10 score for urban older adults was 18.60, which was slightly lower than that for rural older adults (19.43). The mean duration of physical activity per week for older adults was 434.16 min. It was longer for urban older adults (483.83 min) than for rural older adults (325.10 min). Additional information on the characteristics of the participants is shown in Table 1.

### 3.2. Factors Influencing the Level of Physical Activity among Older Adults

The findings in Table 2 show that high levels of education, having a spouse, older age, high self-rated health, and high personal income significantly increased the levels of physical activity among the older adults (*p* < 0.05). Additionally, having a chronic disease also elevated physical activity levels within the older adult population at the 1% level. The likelihood of engaging in moderate-to-high levels of physical activity was 7.5% greater for male older adults than for females when controlling for other variables. Specifically, in rural areas, the probability of engagement in moderate-to-high levels of physical activity was higher for male older adults than for females, with a marginal effect of 8.3%.

The older adults in urban areas were 9.2% more likely to have a moderate-to-high level of physical activity than those in rural areas. Additionally, there were differences in urban–rural factors that influenced the levels of physical activity among the older adults. The physical activity levels of older adult residents in urban areas were affected by age, self-reported health, disability, and personal income, while those of rural older adult residents were influenced by gender, age, education level, spouse’s status, chronic disease, and personal income.

### 3.3. Equilibrium Test for Propensity Score Matching

In this paper, k-nearest neighbor matching was used as an example, with k = 4, resulting in one-to-four matching. Table 3 displays the changes in each explanatory variable before and after matching. The smaller the standard deviation after matching, the better the matching result. Table 3 depicts a reduction in the difference between the variables before and after matching, as well as a significant decrease in deviation. Therefore, it can be concluded that the matching results were improved.

Figure 1 shows the respective distributions of the full sample, urban sample, and rural sample after matching. The results indicate that the variables shifted from a dispersed state before matching to a more cohesive state following matching. The deviations of each variable were reduced and distributed around 0. These outcomes confirm that the matching results were more dependable and effective.

### 3.4. Equilibrium Test for Propensity Score Matching

As shown in Table 4, the effect of physical activity level on depressive symptoms among older adults is relatively close to the treatment effect under the different matching methods, with a mean treatment effect of 2.549. Nevertheless, there is considerable dissimilarity in the importance of the measures, indicating that the matching outcomes are less robust. Among all the participants, there was a significant difference in the treatment effect only for the k-nearest neighbor matching method. High levels of physical activity decreased total depression scores among older adults by a mean of 2.415 (*p* < 0.05). Conversely, treatment effects did not have any significant impact with respect to the caliper and kernel matching methods (*p* > 0.05). Further stratification based on rural and urban location indicated that the impact of physical activity levels on depressive symptoms among older adults was only discernable among urban older adults. However, a substantial difference in the impact effect remained, with significant effects of 2.012 and 2.612 at the 5% level (*p* < 0.05). This study did not find an influential effect of physical activity level on depressive symptoms among rural older adults (*p* > 0.05). It is apparent that physical activity’s effect on depressive symptoms among older adults varies across different categories of older adults.

## 4. Discussion

This study’s findings indicate a higher incidence of depressive symptoms among older adults living in rural areas as compared to those in urban areas, and this outcome aligns with the results of previous research [25]. Yuan et al. [26] also confirmed this trend in their investigation of the urban–rural variations in depressive symptoms among Chinese older adults, demonstrating that 73.96% of the differences in depressive symptoms could be accounted for by area of residence. Furthermore, exercise-related factors accounted for 17.72% of the differences. In recent years, despite China’s urbanization process reaching the mid-to-late stage, the urban–rural disparity in Chinese society remains prominent. There are stark differences between the living environment, living conditions, and needs fulfillment of urban and rural residents, stemming from the socioeconomic differences between both groups [27,28]. Compared to rural older adults, urban older adults enjoy better social support, economic income, and healthcare facilities, all of which bolster their physical and mental wellbeing [29].

However, the relationship between residence and depressive symptoms is inconclusive in international epidemiology. For instance, a national study conducted in Korea reported that the prevalence of depressive symptoms was 1.29 times greater in urban settings than in rural ones [30]. Similar findings have been observed in Nordic regions, where among older adults, residing in urban areas is associated with higher odds of suffering from psychological distress [31,32]. In North America, this difference was relatively minor [33] or even absent [34]. This may, of course, be attributed to variations in the definitions of urban and rural areas, cultural disparities, and ethnic characteristics. Overall, numerous factors complicate the variability in the association between place of residence and depressive symptoms among older adults. Future research needs to consider sociodemographic factors, as well as the influence of socioeconomic factors on this relationship.

This research explored the correlation between physical activity and depressive symptoms among older adults in rural and urban areas of China. According to the results, a higher level of physical activity has a positive impact on depressive symptoms among older adults. This impact led to a significant decrease—by 2.415—in the depression index score. However, this relationship was only discovered in the full sample and among older adults residing in urban areas. Our findings support those of earlier studies showing that physical activity levels are significantly associated with depressive symptoms among urban older adults but not rural older adults [35]. In addition, a study by Mumba et al. [36] found that the correlation between physical activity and depressive symptoms varied among culturally diverse ethnicities of older adults residing in the United States. This discrepancy may be related to varying positive and negative influences on depressive symptoms depending on the social identities and statuses of older adults. Kong et al. [37] found that physical activity had a unique influence on urban Chinese older adults and that the associated influences were more complicated than those for rural older adults. Furthermore, a sizeable cohort study revealed that regular physical activity exclusively benefits individuals with higher levels of activity [38]. The present study’s results corroborate this finding. Urban older adults tend to have more opportunities to engage in sports and social club activities due to their possession of more time, energy, resources, and accessible surroundings, unlike their rural counterparts. In contrast, older adults residing in rural areas face limitations such as a lack of interest, a scarcity of sports facilities, or a need to devote considerable time to creating activities, ultimately reducing the likelihood of engaging in sports.

From another viewpoint, pinpointing precise protective factors for depressive symptoms among older adult individuals is crucial for creating effective measures for prevention and treatment. Life satisfaction, life goals, self-efficacy, and self-esteem are critical protective factors against depressive symptoms among older adults [39]. Furthermore, urban–rural differences affect physical activity and moderate the link between these protective factors and depressive symptoms. In their research, Liu et al. [40] discovered that support from family, friends, and government programs reduces the risk of depressive symptoms among urban older adults and increases positive aging attitudes. However, rural older adults rely more heavily on family support. A well-developed social infrastructure and an abundance of community activities increase physical activity engagement and reduce feelings of isolation and loneliness among urban older adults, thus positively impacting their mental health. In contrast, rural older adults face greater challenges and have difficulty obtaining support from interpersonal networks, leading to similar issues [41]. Additionally, in rural areas, family support tends to manifest itself more frequently in the form of emotional support and companionship across different generations [42,43], with less involvement in physical activities of moderate-to-high intensity. Therefore, appropriate physical activity can be encouraged in these regions to promote intergenerational support and prevent the onset of depressive symptoms among older adults.

Physical activity can help older adults prevent the onset of depressive symptoms. Especially in rural areas, physical activity can serve as a crucial form of family leisure that facilitates intergenerational support. To achieve this, children and grandchildren can participate in aerobics (e.g., walking, running, cycling, etc.) or mind–body exercises (e.g., tai chi, Pilates, yoga, etc.) with older adults on a regular and long-term basis. This approach to promoting health offers dual benefits and can serve as a mutually beneficial family model. To begin with, children can engage in communication with older adults during exercise. This, in turn, can lead to greater intergenerational support with regard to emotional comfort and caregiving, ultimately producing a positive effect on depressive symptoms among older adults [44,45]. Secondly, regular aerobic exercise, especially mind–body exercise, can enhance the cognitive function [46] and self-efficacy [47] of older adults, leading to positive effects on their mental wellbeing [48]. Nonetheless, interventions should be personalized to correspond to the physical capacity of each individual. Studies have evidenced that low-frequency physical activities have more significant effects on the health of older adults. Exercise frequency should be limited to two to three times per week, with each session lasting no longer than 45 min [49].

### Strengths and Limitations

This research was based on the China Health and Retirement Longitudinal Study (CHARLS), utilizing sample data that are broadly representative of the country. Our study examined the impact of physical activity on depressive symptoms among older adults and evaluated the role of urban–rural disparities in this association. This research contributes to the existing knowledge on depressive symptoms among older Chinese adults and offers specific recommendations for such symptoms’ prevention and corresponding interventions. However, there are limitations to this study. The national survey data were insufficient for establishing a causal relationship between physical activity and depressive symptoms due to this study’s cross-sectional design. The investigation into physical activity among older adults was limited by the large-scale testing approach. The results were gathered through self-reported questionnaires, which may be prone to error due to the respondents’ memory bias. Additionally, the categorization criteria for physical activity intensity levels might have skewed this study’s findings. It is important to note that the CHARLS sample aims to obtain a fair and unbiased representative sample using probability sampling proportional to the population size. However, due to notable differences between urban and rural areas in China’s demographic structure, a significant gap in the number of elderly individuals is evident in the sample. It is crucial to consider this aspect when referring to and applying the relevant findings.

## 5. Conclusions

In the present study, rural older adults had a higher probability of depressive symptoms and lower levels of physical activity compared to those in urban areas. Notably, high-intensity physical activity had a positive impact on depressive symptoms among older adults; however, there were distinctions between residents of urban and rural areas. High levels of physical activity only alleviated depressive symptoms among urban older adults.

Physical activity is a crucial aspect of maintaining a healthy lifestyle and has substantial positive impacts on physiological, psychological, and social factors among older adult populations. However, physical activity interventions must consider the living conditions and environment of older adults and be implemented in a timely manner according to their needs and abilities in order to maximize physical and mental health benefits.

## Figures and Tables

**Figure 1 behavsci-13-00864-f001:**
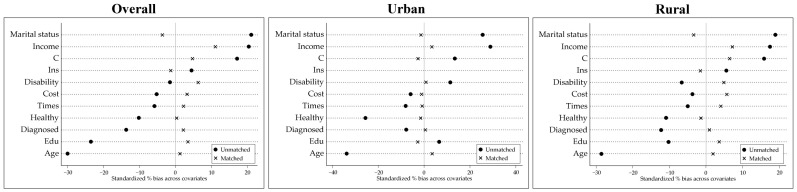
Distribution of control variables before and after matching. Note: Covariates: Edu stands for education; Ins stands for health insurance coverage; Times stands for the number of outpatient visits in a month; and Cost stands for the cost of outpatient visits in a month.

**Table 1 behavsci-13-00864-t001:** Basic characteristics of the sample.

Variable	Variable Description	Overall	Urban	Rural
Sample size	*n*	5055	1582	3473
Gender	Male	48.98%	52.47%	47.39%
	Female	51.02%	47.53%	52.61%
Age	Years	67.55 ± 5.85	67.86 ± 6.18	67.41 ± 5.68
Exercise	mins/week	434.16 ± 364.20	483.83 ± 395.12	325.10 ± 252.70
CESD-10	Scores	19.17 ± 5.26	18.60 ± 4.52	19.43 ± 5.54
Health	Scores	2.89 ± 1.00	2.82 ± 0.94	2.93 ± 1.02
Education level	No formal education	45.52%	25.47%	54.65%
	Elementary school	23.11%	22.19%	23.52%
	Middle school	19.11%	27.31%	15.38%
	High school	10.62%	20.42%	6.16%
	Vocational school and above	1.64%	4.61%	0.29%
Disability	Yes	13.29%	11.00%	14.34%
	No	86.71%	89.00%	85.66%
Marital status	With spouse	83.20%	81.16%	84.13%
	Without spouse	16.80%	18.84%	15.87%
Diagnosed	Yes	43.70%	47.79%	41.84%
	No	56.30%	52.21%	58.16%
Times	<1 time for outpatient treatment/month	92.68%	92.04%	92.97%
	≥1 time for outpatient treatment/month	7.32%	7.96%	7.04%
Cost	ln(spent for outpatient treatment/month)	0.45 ± 1.63	0.51 ± 1.78	0.41 ± 1.55
medical insurance	Yes/medical insurance	97.65%	98.55%	97.24%
	No/medical insurance	2.35%	1.45%	2.76%
Income	ln(any wage and bonus income/year)	1.50 ± 3.40	1.55 ± 3.45	1.27 ± 3.30

Note: The results in the table are percentages/% or means ± standard deviation.

**Table 2 behavsci-13-00864-t002:** Logistic modelling of factors influencing physical activity levels among older adults.

Variable Type	Variable	Overall	Urban	Rural
Covariates	Disability	0.0121	0.4730 *	−0.0710
		(0.0816)	(0.1910)	(0.0879)
	Gender	0.3620 **	0.0627	0.4410 **
		(0.0745)	(0.1780)	(0.0827)
	Edu	−0.1950 **	0.0048	−0.2670 **
		(0.0376)	(0.0726)	(0.0445)
	Area	−1.1360**	-	-
		(0.0918)	-	-
	Marital status	0.3540 **	0.5010	0.3390 **
		(0.1060)	(0.2710)	(0.1160)
	Age	−0.0480 **	−0.0434 **	−0.0498 **
		(0.0065)	(0.0155)	(0.0071)
	Healthy	−0.0773 *	−0.2480 **	−0.0464
		(0.0348)	(0.0830)	(0.0384)
	Diagnosed	−0.1420	−0.0259	−0.1640 *
		(0.0727)	(0.1750)	(0.0803)
	Times	−0.1340	−0.2080	−0.1180
		(0.1130)	(0.2770)	(0.1250)
	Cost	0.0544	0.0655	0.0501
		(0.0553)	(0.1230)	(0.0632)
	Ins	0.3580	0.000	0.2710
		(0.2330)	-	(0.2410)
	Income	0.0276 **	0.0546 *	0.0235 *
		(0.0100)	(0.0217)	(0.0111)
Cons		1.9340 **	0.8890	2.1050 **
		(0.5260)	(1.0630)	(0.5760)
Statistical Testing	N	5055	1559	3473
	R2			
	AIC	5230.3	1058.6	4165.5

Note: Standard are given errors in parentheses. * *p* < 0.05 and ** *p* < 0.01. Covariates: Edu stands for education; Ins stands for health insurance coverage; Times stands for the number of outpatient visits in a month; and Cost stands for the cost of outpatient visits in a month.

**Table 3 behavsci-13-00864-t003:** Changes before and after the matching of propensity scores for each explanatory variable.

Variable	Match	K-NNM (%bias)			*p*-Value		
		Overall	Urban	Rural	Overall	Urban	Rural
Age	U	−30.0	−33.7	−28.5	0.000	0.710	0.650
	M	1.3	3.5	1.9	0.731	0.716	0.636
Gender	U	17.1	13.4	15.7	0.000	-	-
	M	4.7	−2.6	6.4	0.238	0.810	0.140
Edu	U	−23.5	6.4	−10.2	0.000	1.020	0.930
	M	3.5	−2.5	3.6	0.358	0.812	0.393
Disability	U	−1.5	11.4	−6.7	0.640	1.400	0.820
	M	6.3	0.9	4.7	0.097	0.940	0.233
Marital status	U	21.0	25.5	18.9	0.000	-	-
	M	−3.6	−1.3	−3.4	0.302	0.890	0.372
Healthy	U	−10.2	−3.1	−11.0	0.002	0.890	0.900
	M	0.3	94.4	−1.4	0.934	0.894	0.742
Diagnosed	U	−13.7	0.3	−12.3	0.000	-	-
	M	2.2	0.1	0.9	0.578	0.951	0.835
Times	U	−5.9	−8.1	−5.1	0.083	0.630	0.780
	M	2.3	−0.8	4	0.542	0.935	0.309
Cost	U	−5.2	−6.0	−3.7	0.121	0.800	0.850
	M	3.2	−1.2	5.6	0.382	0.913	0.155
Ins	U	4.5	-	5.5	0.188	-	-
	M	−1.3	-	−1.5	0.720	1.000	0.692
Income	U	20.4	28.8	17.4	0.000	1.690	1.310
	M	11.1	3.5	7.1	0.008	0.771	0.116

Note: Covariates: Edu stands for education; Ins stands for health insurance coverage; Times stands for the number of outpatient visits in a month; and Cost stands for the cost of outpatient visits in a month.

**Table 4 behavsci-13-00864-t004:** Influence of physical activity level on depressive symptoms among older adults.

Matching Method	Overall			Urban			Rural		
	Treatment Effect	Standard Error	*p*-Value	Treatment Effect	Standard Error	*p*-Value	Treatment Effect	Standard Error	*p*-Value
K-NNM (*k* = 4)	−2.415	0.078	0.004	−2.012	0.069	0.045	−2.477	0.007	0.055
Caliper Match (caliper = 0.02)	−2.413	0.072	0.058	−2.012	0.065	0.177	−2.477	0.004	0.735
Nuclear Match	−2.819	0.037	0.317	−2.612	0.082	0.038	−2.853	0.043	0.502
Average Value	−2.549			−2.212			−2.602		

## Data Availability

Restrictions apply to the availability of data used. Deidentified participant data are available upon reasonable request made to the corresponding author, with the permission of the CHARLS group at Peking University.

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
