# Peer review of "The Impact of Physical Activity on Depressive Symptoms among Urban and Rural Older Adults: Empirical Study Based on the 2018 CHARLS Database"

_behavsci, 2023, doi:10.3390/bs13100864_

Round 1
Reviewer 1 Report
The aim of the present study is to analyze the relationship between the activity of older adults and the symptoms of depression. A negative relationship has been found, and the difference in physical activity between urban and rural elderly people is also pointed out.
I consider it an interesting and necessary study since the percentage of the elderly population is increasingly higher and it is essential to take care of their physical and psychological health. This will improve their quality of life.
The study is well thought out and developed, but I would like to point out some aspects that may improve the readers' understanding of the study.
In the materials section, when describing the CESD-10 scale, some of the depressive symptoms measured in the scale should be indicated (for example: negative feelings, feeling lonely, not feeling like doing anything,...).
The discussion or conclusions should include a proposal with detailed information on what type of exercise or activities, both moderate and high, and with what frequency, the elderly could do to improve their mental health.
Regarding the sample, it is representative, but the imbalance between the number of elderly belonging to urban and rural areas is striking. This could be commented on in the limitations of the study.
Author Response
Dear reviewer,
Thank you for your efforts in revising our paper. You have suggested detailed, specific and reasonable revisions to improve the quality of our paper. We have improved it according to your revisions. Here are the specifics:
- We have added CESD-10 specific tests in 2.2.1. based on your suggestion.
- We have added relevant recommendations to address the findings of this study at the appropriate place in the discussion based on your suggestions. These are especially for older people in rural areas. The recommendations include the type, frequency and intensity of physical activity and how it should be achieved, suggesting a family model that we call "Healthy Win-Win".
- There is indeed an imbalance in the number of older people in urban and rural areas, which is due to the large regional and urban-rural differences in China's demographic structure. CHARLS uses probability sampling proportional to the size of the population, which ensures an unbiased and representative sample. We have added relevant content to the limitations of the study as you suggested.
Reviewer 2 Report
Please see file attached.

English language needs to be improved. There are issues in word choice, tense use, sentence structure, and consistency.
Author Response
Dear reviewer,
Thank you for your efforts in revising our paper. You have suggested many detailed, specific and reasonable revisions to improve the quality of our paper. We have improved it according to your revisions. Here are the specifics:
- In response to your suggestion that the findings focus on modelling, we have given it some serious thought. Here are some of our ideas that we would like to discuss with you. If you think there are any inaccuracies, we would be happy to revise them. We believe that the presentation of result 3.3 is intended to reflect the process of modelling as well as the degree of applicability. In fact, the propensity score matching model we used in our research methodology was designed to eliminate bias in the observed covariates in order to eliminate the endogeneity of the sample. We provide a more detailed description of the purpose and methodology of model building in 2.3.
- We have made changes to address the repeated errors in the abstract section. This is a problem due to our writing errors. Meanwhile for the measure expression has been optimised.
- Statistical significance: we removed the labelling of P < 0.1 in Tables 2 and 4.
- L.32: We have added the year of China's seventh population census.
- L.35-36: There is an error in the presentation of the individual growth rate. This may be the result of a linguistic error, which we have corrected. In addition, the reference is not to annual growth rates.
- L.91: We have added sampling methods and individual level survey methods as you requested.
- L.102: This refers to the fact that a score of 10 and above is defined as the presence of depressive symptoms, and we have amended the formulation as you requested.
- L.110: We have replaced "paper" with "version".
- L.145: This method of calculation is not specific to China and has been modified in accordance with your request.
- L.211-213: There should be a problem with the language here. What we are trying to say is that the effect has nothing to do with the matching method. We have modified it accordingly.
- Indicator names, spelling, capitalisation and formatting of all tables and figures have been revised accordingly. The corrections should be in line with academic writing standards.
- P-values have been added to Table 4.
- With regard to the formulation of the reference to older adults, we consider your comments to be very important and reasonable. Therefore, we have made the appropriate changes in accordance with your comments.
- L.78: We have replaced "research" with "study".
- L.86: We have amended this sentence to address the grammatical error that appeared in it.
- L.104: We have replaced "analyse" with "collects".
- L.128-129: We have made changes to address the capitalisation of the name of the questionnaire.
- L.165: We removed this sentence left by the template. This was a problem when applying the template for copying.
- L.182-187: we have made changes to address the tense issues in this paragraph.
- L.237: There was a problem with the editing of this sentence, which has been corrected.
- In response to the more problematic word choice, sentence structure, and tense of the English language, we have duly chosen MDPI's language editing service to assist us in checking for possible problems elsewhere, which we may not have found very easily. As a result, we believe that the English presentation of this draft should be much improved.
Reviewer 3 Report
Dear Authors
I congratulate you on your work. It would be interesting to perhaps add a hypothesis about what you intend to look for in the article.
The presentation of the methodological part helps a lot to understand what is going to be presented next.
Best regards
Author Response
Dear reviewer,
Thank you for your efforts in revising our paper. You have suggested detailed, specific and reasonable revisions to improve the quality of our paper. We have improved it according to your revisions. Here are the specifics:
- We have formulated our research hypothesis in the appropriate place in the preface section in accordance with your comments. And based on such formal requirements, we have adjusted the rest of the foreword.